# Journey of the Probiotic Bacteria: Survival of the Fittest

**DOI:** 10.3390/microorganisms11010095

**Published:** 2022-12-30

**Authors:** Allyson Andrade Mendonça, Walter de Paula Pinto-Neto, Giselle Alves da Paixão, Dayane da Silva Santos, Marcos Antonio De Morais, Rafael Barros De Souza

**Affiliations:** 1Laboratory of Microbial Genetics, Department of Genetics, Federal University of Pernambuco, Recife 50670-901, Brazil; 2Laboratory of Microbial Metabolism, Institute of Biological Sciences, University of Pernambuco, Recife 50100-130, Brazil

**Keywords:** acid stress, atomisation, carbon balance, central carbon pathway, stress tolerance

## Abstract

This review aims to bring a more general view of the technological and biological challenges regarding production and use of probiotic bacteria in promoting human health. After a brief description of the current concepts, the challenges for the production at an industrial level are presented from the physiology of the central metabolism to the ability to face the main forms of stress in the industrial process. Once produced, these cells are processed to be commercialized in suspension or dried forms or added to food matrices. At this stage, the maintenance of cell viability and vitality is of paramount for the quality of the product. Powder products requires the development of strategies that ensure the integrity of components and cellular functions that allow complete recovery of cells at the time of consumption. Finally, once consumed, probiotic cells must face a very powerful set of physicochemical mechanisms within the body, which include enzymes, antibacterial molecules and sudden changes in pH. Understanding the action of these agents and the induction of cellular tolerance mechanisms is fundamental for the selection of increasingly efficient strains in order to survive from production to colonization of the intestinal tract and to promote the desired health benefits.

## 1. Probiotics: The Current State of an Ancient Practice

Probiotics have been used for more than 10,000 years by humans [1]. The use of these microorganisms, unknown for the great majority of this period, occurred in many cultures for the production of fermented beverages [2]. Nakazawa and Hosono [3] reported that early food manufacturers used bacteria and yeasts without knowledge of their existence to produce fermented dairy products, some of which being produced as early as 3500 BCE, and they are still common in the Middle East [4].

The agents of fermentation were first reported by Louis Pasteur in the decade of 1850 when he established yeasts as living beings and described the lactic acid-producing bacteria (LAB). Henry Tissier, a paediatrician at the Pasteur Institute, was the first to relate the use of bacteria to the treatment of intestinal diseases in 1906 [5]. A year later, the Russian microbiologist Elie Metchnikoff, in agreement with the Bulgarian physician Stamen Grigorov, introduced probiotics as agents of longevity of life in his book “*The Prolongation of Life: Optimistic Studies*”. In that book, Metchnikoff suggested that the ingestion of some specific bacteria may beneficially influence the gastrointestinal tract (GIT) of humans. This hypothesis was based on the observation of the relationship between the long and healthy life of Bulgarian peasants and the consumption of large amounts of fermented dairy products [6]. Since then, probiotic microorganisms have been isolated from the most varied substrates and currently applied for a myriad of purposes beyond infectious diseases of the GIT, at a moment in history that Ozen and Dinleyici [1] called this period as the “Age of Probiotics”.

From here, we will explore this world of probiotics from their production to their final destination, evaluating points that are relevant to their physiology, the challenges that cells face from the factory to their final destination in the consumer, and discussing some of their actions beneficial (Figure 1).

### 1.1. The Origin: Evolution of Probiotic Meaning

The term “probiotic” was first coined to designate substances produced by bacteria or yeast that promoted the growth of other microorganisms [7]. Since then, this concept is constantly under revision and is updated to cope with the evolution in food technology and microbiology with new definitions being reported over the decades (Table 1). Along the concept evolution, the improvement of intestinal health was pivotal to any probiotic definition. Although, the main change over the years was due to the increasing attention to the organism instead its metabolites.

Currently, the most widely used definition is the one proposed by the Food and Agriculture Organization of the United Nations (FAO) and the World Health Organization (WHO), that was updated by the International Scientific Association for Probiotics and Prebiotics [13,16]. In this definition, a probiotic is a live microorganism that improves the host’s health when administrated in adequate amounts. However, recently Zendeboodi et al. [15] proposed a new design for probiotic products that deviates from previous concepts. In this new formulation, non-viable or even disrupted probiotic cells can have beneficial effects on the host [15]. Additionally, several new terms emerged in the literature to handle the innovations in the production, formulation, effectiveness and safety of those products (Table 2).

Despite the current innovation in the field, new applications and research are in development that could ensure future reconceptualization. For instance, research in recent decades have expanded the application of the probiotic beyond health promotion but also including health restoration in ill hosts. In this scenario, medical formulation of those microorganisms can be employed for therapeutic purposes to treat gastrointestinal tract (GIT) distresses from infectious or non-infectious cause [17,18]. Since there is no restriction on probiotic consumption, this could be a safe way to treat new diseases without a proper pharmacological approach, such as in the beginning of the COVID-19 pandemic [18]. Additionally, any propriety of a probiotic can be enhanced by gene engineering to solve economic issues or to adjust to new purposes [19]. For the purpose of this review, we will focus only on natural unmodified probiotics.
microorganisms-11-00095-t002_Table 2Table 2Glossary containing the main terminologies associated with probiotics in the specific literature.NameDefinitionReferencesParaprobiotic“Dead bacterial cells or cell components to denote health benefits beyond the inherent viability of probiotics.”[20]PostbioticAny substance released by or produced through the metabolic activity of the microorganism, which exerts a beneficial effect on the host, directly or indirectly.[21]Prebiotics“Substrate that is selectively utilized by host microorganisms conferring a health benefit.”[16]SynbioticMixture of prebiotics and probiotics for the improvement of human or animal health.[22]


### 1.2. Screening and Selection: The Search for the Best

The search for new probiotic strains has been stimulated in recent decades due the interest in healthier and functional foods [23,24]. Additionally, the concern of synthetic drugs in allopathic medicine and the rise of bacterial resistance, has increased the pursuit for non-pharmacological treatments such as the administration of probiotics [25,26,27,28,29]. Currently, the use of those microorganisms is considered a suitable alternative for the treatment of various infectious and non-infectious diseases and even metabolic disorders [17,18,22]. Despite any application, a given strain must attend some functional and safety criteria to be considered a probiotic. Based on these, it is important that new strains be sought and tested to ensure its safety, which is usually stated by the status of “Generally Recognized as Safe” (GRAS). However, the GRAS status alone is not sufficient to be a probiotic. It is also required that the strain colonizes the GIT and promotes some health improvement to the host. To fulfil this requirement, those strains have been isolated from animal or human GIT samples and stool to ensure effectiveness in colonization. Probiotic strains have been isolated even from human milk, which was associated with a good health condition of infants [30]. Although, there is no restriction of isolation source if the strain is not virulent or pathogenic [6]. Genomic analysis has revealed interesting insights on the discrimination between probiotic and pathogenic bacteria based on the genes involved in cell mobility and mobile elements [31] and in the synthesis of O-antigens [32]. After isolation and identification, the microorganism must proceed to functional tests in vitro and, later, in vivo, according to FAO/WHO guidelines [13].

The main in vitro tests used for screening of probiotic are the following: (1) resistance to gastric acidity; (2) resistance to bile acids; (3) adherence to mucus and/or human epithelial cells and cell lines; (4) antimicrobial activity against potentially pathogenic bacteria; (5) ability to reduce the adhesion of pathogens to surfaces; (6) bile acids hydrolase activity [13]. In addition, some probiotic applications may require strains with specific features, such as the resistance to spermicides for probiotics for vaginal use [33]. Other proprieties include cholesterol-lowering capacity, antioxidant activity, cytotoxic effect against cancer cells, bioavailability of vitamins and minerals, modulation of immune response, intestinal motility, anticaries activity and others [34,35,36,37]. However, the heterogeneity of diet, age and microbiome in the human, imposes a challenge to the identification of new probiotic that exerts its beneficial effect in all hosts [38].

Most of the known probiotic species belong to the Lactic Acid Bacteria group (LAB), such as the lactobacilli group which had its classification recently revised [39]. Others probiotic strains that hold economic interests includes species from the genus *Bifidobacterium* [40]. The main bacterial species of the genera Lactobacillus and Bifidobacterium used as probiotics are *L. acidophilus, L. casei, L. crispatus, L. delbrueckii subsp. bulgaricus, L. fermentum, L. gasseri, L. helveticus, L. johnsonii, L. lactis, L. paracasei, L. plantarum, L. reuteri, L. rhamnosus, L. sporogenes*, *L. acidophilus, L. casei, L. crispatus, L. delbrueckii subsp. bulgaricus, L. fermentum, L. gasseri, L. helveticus, L. johnsonii, L. lactis, L. paracasei, L. plantarum, L. reuteri, L. rhamnosus, L. sporogenes* [41,42]. Noteworthy, the probiotic status can also be assigned to yeast species, such as those belonging to the *Saccharomyces* senso strictu group, such as *S. cerevisiae* subsp. *boulardii*, although most of the probiotic strain available in the market are bacteria [43]. This review will be restricted to bacterial probiotics. In addition, we will continue using the old classification of bacterial species because they have a strong connection with products already available on the market.

### 1.3. Applications in Contemporary Times: The Search for “Live Long and Prosper”

The search for food that not only provides nutrients but also benefits to the health of the consumer, even ameliorate disease conditions, have brought the concept of nutraceuticals. Nutraceutical products include the use of these fermented foods as anti-pathogenic, anti-diabetic, anti-inflammatory, anti-cancer, anti-allergic, angiogenic, and urogenital, brain and central nervous system (CNS) health care activities [44]. Administration of *L. fermentum* in high-fat-fed rats alleviated some metabolic disorders that are induced by this type of diet [45]. In addition, the oral administration of *L. rhamnosus* LPR strain promoted accelerated skin wound closure in mice, which opens therapeutical approaches for skin wound treatment [46]. In a very recent article, Li et al. [47] reported the positive effects of *Lactobacillus reuteri* in protecting elderly women from osteoporosis. Other nutraceutical proprieties of the probiotics include reduction of cholesterol and triglycerides and the prevention and treatment of diabetes and obesity [48,49,50,51]. These therapeutic aspects are also of interest to the pharmaceutical and food industry, as they can be used as a marketing or branding strategy that has increasingly attracted the attention of the consumer public [52]. Nascimento et al. [53] proposed the formulation of a nutraceutical containing *L. fermentum* 296, quercetin and resveratrol which maintained high bacterial activity over 90 days of storage. Despite the various metabolic benefits that probiotics have demonstrated for the health of consumers, this is an issue that must be evaluated very carefully. For that matter, the reader can consult the recent review by Kim et al. [54] that specifically reported little evidence of the benefits of probiotics, but also prebiotics and synbiotics on the incidence of colorectal adenomas and cancer. In the same line of reasoning, Thangaleela et al. [55] argue that there is even evidence of the benefits of probiotics in the improvement of patients with neurological disorders, such as Alzheimer’s and Parkinson’s diseases. However, the authors alert to the fact that these benefits depend on several variables, from the type of probiotic strain to the patient’s physiology. This opens up a very promising field of research regarding the search for personalized treatments.

In addition to consumption by humans and their benefits, the use of probiotics in the supplementation of livestock feed, such as poultry, swine and ruminants, has also been gaining importance. Several studies have shown that much of the benefits attributed to probiotics in humans are also found in these animals, and this has made it possible to raise healthier animals, with less use of drugs or equivalents and with a decrease in fattening time [6,56,57,58]. The production and marketing processes of probiotics bear much similarity for human and animal use, with cells facing very similar metabolic challenges. However, when it comes to consumption, this article will focus on the main metabolic characteristics of probiotic bacteria during their trajectory in the human GIT.

Another very interesting application of probiotics is in the formation of a protective film for the preservation of fruits and vegetables from the antagonistic activity of these bacteria against spoiling microorganisms, such as fungi and bacteria [59]. The application of this technology based on the preparation of edible films in which the bacteria are fixed can constitute an important step towards the reduction of chemical products used for the preservation of these foods.

### 1.4. The Market: Opportunities beyond the Big Players

The growing market demand for natural methods capable of performing positive functions in human health has intensified the search for probiotic products and, consequently, their production on an industrial scale [40,60,61]. The global probiotics market in 2021 reached the value of USD 58.2 billion, with an estimated increase at a compound annual growth rate (CAGR) of 7.5% until 2030, yielding a forecast revenue of USD 111.2 billion [40]. The probiotic market can be divided into a few segments, which include items such as probiotic drinks and foods, dietary supplements and probiotics used in animal feed. Distribution channels include super/hypermarkets, pharmacies, specialized and online stores [62]. Probiotic foods and beverages include dairy, non-dairy products, probiotics for dry foods, among others and have reached the top of the global market with over 75% revenue [40]. This growing interest is in part triggered by the recent development of new applications, consumers’ new habits and the diversity of food matrices such as yogurts, fermented milks, beverages, dairy desserts, cheeses, kefir, sauerkraut and juices that diversifies the offering of the product. Additionally, some products are an alternative for the population with dietary restrictions, such as lactose intolerance, allergy to milk proteins and hypercholesterolemia [63,64,65,66,67,68]. To attend the growing demand, the challenge of the food industry is to widen the diversity of strains used for different purpose and to scale up the production of these bacteria biomass.

In addition, the great demand for animal protein sources and the increase in industrialized livestock favoured the animal feed market, which obtained revenue of US $4.2 million in 2018, with an expected CAGR of 8.8% by 2024 [62]. The markets, which hold the largest share of sales in the world, obtained revenues of more than 30%. Most of the probiotics marketed are of bacterial origin and, for this reason, the segment had revenues greater than 80% [40].

Currently, Asia Pacific is emerging in the global market for probiotic products, generating revenue of more than 40% in 2021, due to the influence of the media and public awareness. Although the increase in the elderly population has influenced the increased consumption of probiotic products, “millennials” stand out as the largest consumers [40,62]. China and Japan are expanding their market in the region. In Japan, a high demand of probiotics supplements is expected due to the large number of elderly people in the country. In China, high meat consumption drives the market for probiotics incorporated in animal feed [69]. In 2019, the Brazilian market for probiotics was valued at USD 1.5 million and there is the expectative for significant growth until 2025 [70].

The presence of large market players such as General Mills, Chr. Hansen, Danone, BioGaia and Lallemand Inc., etc., favours the consolidation of the domestic market, as they expand and develop new products [62]. However, more and more the people are searching for non-industrialised artisanal products, which makes a profitable market for small farmers and small producers all over the world. Small companies have emerged to produce and supply probiotics to be added in the preparation of juices, beverages, dairy products, bakery products, etc., creating a very interesting and attractive market for new entrepreneurs. So, the market is open to different opportunities, from small companies in the cities that produce the cells to enterprises in the cities (bakeries and natural, vegetarian and vegan food stores, for example) and in rural areas (small farms) that produce dairy products. The challenge for these producers focuses on production efficiency combined with the delivery of cells with high viability and vitality so that they can fully exercise their probiotic function. Next, various biological aspects related to these challenges will be discussed.

## 2. From the Physiology of Production to the Processing of the Product

The industrial improvements brought by the increasing in probiotic consumption includes the increment of the bacterial biomass and the stability of the cells to conserve its vigour during the product shelf-life. Culture purity is a critical issue because contaminant bacterial or yeast can produce undesirable molecules that induced the loss of probiotic cell viability in the product [71,72]. Several methods are available to control the contamination such as pasteurization. Regarding to the inoculum, cultivation on a small scale in the laboratory can ensure the culture purity before any scale-up application.

Most of probiotic lactobacilli species are considered oxygen-tolerant anaerobes that do not present a complete electron transport chain (ETC), whose aerobic growth is dependent on the presence of electron acceptors that make ETC minimally active [73]. The presence of oxygen might induce the production of reactive oxygen species (ROS) that leads to damage to the cellular components, such as proteins, lipids, and nucleic acids [74]. The best bacterial growth performers in the presence of oxygen are those strains that are capable of producing oxidative stress protective agents such as catalase and the electron acceptor molecules [75,76]. One strategy is to add such molecules in the substrate for aerobic cultivation, if they do not represent high costs for the production process. To repair the damaged molecules, the cell needs to expend more energy in reparative mechanisms or even in the synthesis of new components. On the other hand, cultivations in anaerobic conditions induce the fermentative metabolism that dissimilate carbon from the substrate in the form of fermentation products, such as lactic acid. Thus, the more carbon is lost, the less it is assimilated to bacterial biomass [77]. Therefore, the fate of the assimilated carbon defines the yield of biomass formation, or simply biomass yield, calculated as the mass of the cell produced by the mass of carbon consumed. Pedersen et al. [78] showed how the oxidative metabolism using respiration can increase biomass production and yield and how it can produce cells more tolerant to different forms of stresses. Hence, any application that demands high cellular viability can be achieved by keeping the oxygen at a minimal level enough to maximise biomass yield and to prevent oxidative stress [75]. Therefore, the control of the contamination and the oxygen level in the batches are straightforward parameters for any industrial process for probiotic production. The temperature of fermentation may also be considered, although most of the LAB species have a wide optimal growth temperature between 30 to 37 °C. Species isolated from mammal samples more likely has better growth at normal body temperature of 37 °C. The appropriate temperature fastens the bacterial growth and the biomass production for process that requires high cellular density. Proper cell density input in the fermentative process saves time, reduces the cost, and imposes a competitive barrier for spoiling microorganisms to establish.

In addition to their inherent probiotic proprieties, the selected strains might be of easier growth ability to produce high cell density cultures in different substrate or production matrices. Alternatively, improvement of the production processes can be achieved by supplementing the substrates with growth factors or by mixing different substrates. The availability of nutrients must influence in cell metabolism and, consequently, in the efficiency of biomass formation. These nutrients are metabolised throughout the so-called central metabolism that remove energy in the form of ATP and reduced co-factors and distribute the carbon to the different metabolic pathways to produce building blocks for cell components. However, during metabolic transformations, part of the carbon and energy can be deviated to produce metabolites that, in the context of this industry, are considered by-products, such as organic acids and alcohols. The more these metabolites are produced, the less carbon is assimilated to biomass. On the other hand, these metabolites are also part of the whole probiotic formulation when considering cell suspensions or fermented dairy, which include the viable cells (probiotics) and their fermentation products (postbiotics). Thus, the challenge is to establish ideal fermentative processes in which the maximal of cell growth and biomass yield are in equilibrium with the cell metabolism and stress tolerance.

### 2.1. Uptake of Sugars: The Sweety Diversity

The production of probiotic biomass is the step that may represents the highest cost, since LAB are very fastidious microorganisms. It depends on the presence of a sugar, organic sources of nitrogen and growth factors [79]. Thus, the biomass yield will depend on the nutrients available in the production substrate as well as on the efficiency of the central metabolism of the cultivated bacteria. In this context, it is important to understand how LAB can use the nutrients available in the substrate and what metabolic challenges are encountered.

To initiates the metabolization of any nutrient the cell needs to uptake them from the medium to the cytosol where the bioconverting enzymes resides. Some nutrients can enter the cell spontaneously by a gradient concentration whenever the cell envelope does not impose any restriction. Many other components, however, cross the cell envelop through a series of transporters and channels [80]. Hydrophobic substances can cross the cell envelope with minor resistance because the lipid bi-layer membrane [81]. On the other hand, most of the nutrients required by the cell are hydrophilic and incapable to cross the membrane without assistance. In general, this assisted transport spends energy reserve by consuming adenosine triphosphate (ATP), the energy currency of all living beings [82,83]. Despite the diversity of protein specialized in molecule transport across the cell membrane, the majority fall into three main mechanisms that allow the cell to concentrate compounds internally against their concentration gradient: (1) the ATP-Binding Cassette (ABC) transporters, (2) the Major Facilitator Superfamily (MFS) transporter and (3) the PhosphoTransferase System (PTS). While the ABC transporter and MFS transporters are found in all domains of life, PTS is exclusive of prokaryotes [83]. The presence of these transporters shows the effectiveness of a given strain in the uptake of different prebiotics sugars, and it defines their potential as probiotics.

The ABC transporters are composed of several proteins that altogether can interacts with the sugar, the cell membrane and ATP molecule during the transportation steps. The external portion of an ABC transporter that uptakes sugar is composed of the Sugar Binding Protein (SBP). In Gram-positive bacteria, this SBP is usually a specialized domain from the Transmembrane Protein Subunit/Domain (TMD) or a separated protein anchored elsewhere in the membrane [83]. The transport process initiates with the formation of sugar-SBP complex that interact with the TMD, leading to a transition state that will unblock the channel into the cell. Along this process, the internal portion of the ABC transport binds ATP by the Nucleotide Bind Domain (NBD) [83]. In this step, the sugar is transferred from the SBP to the TMD. Afterwards, ATP binds to NBD to induce conformation changes at the TMD to push the sugar into the cytosol. Lastly, ATP is hydrolysed by NBD and TMD returns to its original state to initiate a new transportation cycle [83]. This transport system has been associated with the uptake by the probiotic species of several prebiotics such as inulin (a fructose polymer), galactooligosacharide (GOS), raffinose (glucose + fructose + galactose), panose (a glucose trimer), xylobiose (a xylose dimer) and xylooligossacharide (XOS) and other sugars [82,84,85,86]. The lack of this transport can influence in the production process that use substrates enriched in some of these sugars, as well as reduce the fitness of probiotic during colonization of GIT, mainly in neonates that have a diet composed of their mother’s milk with several oligosaccharides [87]. Regarding to the MFS, this transport system moves one molecule against its concentration gradient by exploiting the transportation of a second molecule in favour of its concentration gradient [83]. This transport does not demand ATP directly [83]. Many MFS transporters employing the movement of ions across the membrane to transport nutrient to the cell or to export toxic compound from the cytosol [88]. However, the increasing intracellular concentration of that ion demands an active transport for its extrusion outside the cells at expenses of ATP. This transport mechanism can also be employed by the probiotic strains to extrude toxic compound found in the GIT, such as bile acids and lactic acid [89].

Although ABC and MFS transports can be employed by the probiotic cells, PTS seems the more energetically efficient mechanism to transport sugars [83]. It employs a mechanism that involves the use of specific protein transporters that internalize sugars concomitantly with their phosphorylation with the use of phosphoenolpyruvate (Pep) from the glycolysis as phosphate donor [86]. These systems are used for the transport of glucose, fructose, sucrose (glucose + fructose), maltose (a glucose α-dimer) and cellobiose (a glucose β-dimer), and they are quite relevant if the cells are to be cultivated on industrial substrates such as corn syrup, sugarcane juice, fruit juices and cellulose hydrolysates. The number of genes encoding PTS transporters can vary greatly in LAB, even varying between individuals of the same species [90]. In the case of disaccharides, only one of the monomers is phosphorylated during the transportation by the PTS [88,91]. When internalized, they are cleaved by the corresponding phosphoglucohydrolases and the non-phosphorylated monomer is afterwards phosphorylated by the internal kinases [92,93,94]. This diversity of carriers can have relevant consequences in the process of choosing the best production substrate for the cultivation of the selected probiotic to increase biomass production and cellular viability. Thus, sugar transport and metabolization is critical point to any industrial fermentative process.

### 2.2. The LAB Biochemistry: Always Lactic, Sometimes Ethanoacetic

Sugar, as in other living beings, serve as source of biomass carbon and energy, although it may not be used by LAB as the main source of carbon [94,95]. There are different ways that the assimilated carbon can take the Central Carbon Pathway (CCP) (Figure 2). Mono and oligosaccharides are processed and converted to glucose 6-P (Glu-6P), Fructose-6P (Fru-6P) and/or glyceraldehyde 3-phosphate (G3P) for their metabolization to biomass components or to fermentation products. Galactose from milk and fruits has to be first metabolised via the Leloir pathway to ends up as Glu-6P, while mannose that is also extracted from plant fibres is first phosphorylated to mannose 6-phosphate (Man-6P) and then is epimerised to Glu-6P. Fructose enters as fructose 6-phosphate directly in the glycolytic pathway.

Despite its origins, Glu-6P is further isomerized to Fructose-6P (Fru-6P), phosphorylated to fructose-1,6 biphosphate (Fru-1,6P) and breakdown into two triose phosphates: glyceraldehyde 3-phosphate (G3P) and dihydroxycetone 3-phosphate (DHAP). Each mol of DHAP is isomerised to G3P and both G3P molecules follows the oxidative part of glycolysis for the final production of 2 mols of pyruvate, 2 mols of NADH and 2 mols of ATP. Alternatively, Glu-6P goes to the Pentose Phosphate Pathway (PPP) through its oxidation to phosphogluconate and ultimately by decarboxylation to ribulose-5 phosphate (Ribu-5P), with the production of two reducing equivalent in the form of NADPH [94,96]. Then, part of Ribu-5P is isomerised to ribose 5-P (Rib-5P) and part is epimerised to xylulose-5 phosphate (Xylu-5P). A small fraction of Rib-5p is deviated for nucleotide biosynthesis while the majority of Rib-5P molecules condensates with xylu-5P for a series of transaldolase and transketolase reactions in the PPP [96,97,98]. At the end, this pathway provides Fru-6P and G3P, which return to glycolysis (Figure 2), and erythrose-4P (Ert-4P) used for amino acid biosynthesis [99].

In some lactobacilli, an alternative carbon catabolic pathway evolved to merge glycolysis and PPP in a single route [94,96] (Figure 2). In this pathway, Glu-6P is converted to Xyl-5P through the oxidative part of PPP, with the generation of NADPH, followed by the breakdown of Xyl-5P into G3P and acetyl-CoA by the enzyme phosphoketolase (PK) [100]. G3P enter the oxidative part of glycolysis for ATP and NADH production, while the fate of acetyl-CoA will depend on the redox state of the cytosol [101]. Acetyl-CoA can exchange CoA by inorganic phosphate (Pi) if the cells are capable of regenerating NADPH produced by PPP using any electron acceptor molecule [102]. The produced acetyl-P is dephosphorylated by acetate kinase (Ack) to produce acetate and ATP. This combined PPP-PK pathway provides the physiological capability also to assimilate pentoses (Figure 2). For this, xylose and arabinose are isomerised to xylulose that is further phosphorylated to Xyl-5P, the substrate for PK [103,104,105]. The metabolization of the pentoses from dietary fibres such as hemicellulose improves the production of short fatty chain acid (SFCA) from acetyl-CoA produced by PK activity, and these SFCA that can be assimilated by the gut epithelial cells as postbiotic product [106]. Some studies also suggest the PPP plays an important role in the tolerance to acid stress in *L. reuteri* [107] and during the growth of *Bifidobacterium* on the prebiotic XOS and β-glucans [108,109].

The metabolism of sugars by these pathways described above can induce different metabolic states of probiotic cells, which led to the classification of LAB species into three metabolic groups based on the profile of products associated with cell growth (Figure 3). The first group corresponds to the homofermentative bacteria that have lactic acid as the major or unique metabolite [92,96,110]. This is due to the cell’s need to re-oxidise the NADH produced in the oxidative part of the glycolytic pathway by reducing pyruvate to lactic acid [96] (Figure 3a). In this case, hexoses would serve more as a generator of ATP at the substrate level in the oxidative phase of glycolysis since much of the carbon would be dissimilated as lactic acid that is extrude from the cell. In this case, the highest industrial biomass yield would depend on the presence of other carbon sources, such as peptides and amino acids, citrate and acetate that can provide intermediates for biosynthetic reaction rather than energy for ATP synthesis [102,111]. Thus, the production of a homofermentative probiotic from sugarcane juice or corn syrup, the cheapest sugar-containing industrial substrates on the planet, would most certainly require supplementation with these carbon-supplying molecules. This fastidiousness of probiotics can make the production step quite expensive. The alternative is to aerate the production tanks, so that the oxidative metabolism can do the task of NADH re-oxidation, leaving pyruvate available for anabolic reactions. In this case, the first concern should be with the evaluation of the process aeration cost versus the substrate supplementation cost. The second concern is the need to produce complete products that contain cells (probiotic) and their fermentation products (postbiotic), inhibited the more oxidative growth there is. Therefore, the search for cheap substrates that fulfil the nutritional requirements, if they are allowed for food production, is a very relevant research topic. In this context, the processing of protein-rich substrates such as corn steep liquor and soybean extracts seems attractive.

The species that employs the PK reaction constitutively are classified into the second group that corresponds to the obligate heterofermentative bacteria (Figure 3b), with a diverse profile of metabolites produced during growth in the presence of hexoses or pentoses, which include lactic acid, CO_2_, acetic acid and ethanol [79,112]. This is due to the presence of the alternative PPP-PK metabolic pathway described above. In these bacteria, both hexoses and pentoses are obligatory converted by PPP to Xyl-5P that is metabolized via PK to produce lactic and acetic acid in redox balanced condition [79,91,105,112]. In unbalanced redox conditions, ethanol is also produced to regenerated NAD^+^ [110,113]. As in the first type, the fate of the carbon will define the yield of biomass, since the more fermentation products the less cell components are synthesized.

Some species can use the PK reaction without the oxidative phase of PPP sole to metabolize pentoses and composes the third group of classification called facultative heterofermentative (Figure 3c), which is homofermentive for hexoses and heterofermentive for pentoses [110]. There is also the very peculiar and, so far, exclusive case of the *Lactobacillus vini* species, which is homofermentive for both hexoses and pentoses [111,114]. Some LAB species were recently classified as fructophilic (BALF) [115,116]. In these bacteria, fructose serves as an electron acceptor and the cells produce mannitol to restore the redox balance [96,110,115]. Therefore, the efficiency of biomass production of these bacteria should be higher in sucrose than in hexoses, since glucose could provide carbon for anabolic reactions while fructose could balance the redox state. In this case, the use of sucrose-rich substrates such as sugarcane juice or molasses, or even sweet sorghum, would be very suitable for industrial production of these probiotics. 

Glu-6P, that represents the starting point of these different metabolic states, is also the first metabolic crossroad of the CCP. The major fraction flows towards glycolysis and/or PPP-PK pathway, but a small part of this is converted to glucose 1-phophate (Glu-1P) for the biosynthesis of the storage carbohydrate glycogen. It accumulates in some probiotic species depending on the carbon source and the growth phase [117]. For example, growth on trehalose induces glycogen synthesis in early log phase of *L. acidophilus*, but that glycogen is almost entirely consumed before the entrance in the stationary phase [117]. On the other hand, the glycogen stock is depleted in the early log phase on raffinose in this species, but not re-filled in the stationary phase [117]. In this bacterium, the glycogen biosynthetic genes are organized in an operon that is negatively controlled by glucose through the Catabolic Repression Elements in its promoter region [118]. This regulation suggests that accumulation of glycogen occurs when the concentration of glucose decreases below a critical threshold that weaken the catabolic repression. Since glycogen might protect some LAB species from environmental stresses (see below), the industrial production of glycogen-rich LAB biomass seems advisable both for the processing and storage steps and for survival in the GIT of the consumers. Another important biosynthetic pathway initiates with Fru-6P from the CCP to the bacterial cell wall biosynthesis [119,120,121]. This process starts when Fru-6P molecules are aminated and acetylated in a series of reactions to produce the N-acetylglucosamine (N-AcGlu). Then, part of this intermediate is reduced by NADH to N-acetylmuramic acid (N-AcMu). Afterwards, N-AcGlu and N-AcMu are covalently linked through a β-1,4 glycosidic bound to form the building block of the bacterial cell wall [120,121,122]. The correct structure and stability of the bacterial cell wall is of paramount importance for the quality of the probiotic product and its action in the GIT.

Whatever the type of metabolism (Figure 3), LAB will always produce lactic acid as a metabolite associated with bacterial growth. However, this weak organic acid can affect biomass yield due to a kind of self-imposed stress [123]. In its protonated form (at medium pH below 4), lactic acid crosses the cell membrane by simple diffusion [124,125]. Inside the cell, this molecule dissociates into the lactate anion and the H^+^ proton, which leads to acidification of the cytoplasm [126]. Like oxidative stress, acid stress produces direct damage to cellular structures and imposes an energetic burden through the action of ATP consuming extrusion pumps [127]. This seems to be less relevant in complex substrates due to the presence of buffering agents such as amino acids, but it is relevant when considering the use of simple industrial substrates. Some studies point to lactic acid also as a metabolic regulator capable of inducing growth arrest and early entry into the stationary phase [128,129]. All these factors can also affect the biomass yield and decrease industrial production, with consequent raising of the production costs.

At the end of the cultivation stage, the cells can be either used for direct consumption, as cell suspensions, or separated from the medium through filtration or centrifugation approaches, concentrated and subjected to drying processes (lyophilization or atomization) for later incorporation into food matrices or sale in the form of powder in capsules, sachets, among others. To avoid the loss of cell viability inherent to the drying processes, some protective agents can be added in advance [130].

### 2.3. Processing Probiotic Cells: Dried but Not Dead

Probiotics in dry form have several applications and advantages over liquid suspensions, such as greater stability and easier storage and transport, which do not require special structures such as refrigeration [131]. Several pharmaceutical forms can be based on dried cells such as capsules, tablets, and sachets for oral administration [63,132,133,134,135]. They can also serve as a starter culture for application in the dairy industry and as food supplements [136,137,138]. For these purposes, the main techniques for drying probiotic cell in the industry include freeze-drying and spray-drying [139].

Currently, freeze-drying (lyophilisation) is the most industrially used method as it removes water without the application of high temperatures. It preserves the non-thermotolerant microorganisms and provides greater cellular viability in the end of formulated products [140,141,142]. In the dried state, the metabolic activities of the cell and the biological functions are paused but are recovered after rehydration [131]. This method basically consists of three steps: freezing the sample between −20 to −80 °C, followed by primary drying in which the external liquid of the sample is slowly sublimed [131,143]. Then, the internal liquid is the removed with a secondary drying by the method of desorption [131,143,144,145]. During sublimation, the suspension must be properly frozen to ensure greater viability. If the temperatures applied are not properly low, the suspension will not freeze completely, which implicates in final viability losses. On the other hand, if the temperature is excessively low the cell integrity can be compromised [144,146]. The need to pre-freeze the probiotic suspension generates high energy consumption and limits large-scale production [142,147].

Because it is a slow process, lasting between 24 to 48 h, it can favour the appearance of ice crystals that may cause cell membrane damage and protein denaturation. Damages to membrane lipids and cellular proteins also occur because of oxidative stress. Beyond thermal stress, osmotic stress is also generated by the drying process, due the decrease in the solubility of the liquid surrounding the cell. It leads to variation in the concentration of solids in the suspension, increasing the osmolarity and favouring the passage of water through the cell membrane, which could lead to plasmolysis [148,149,150]. Therefore, some strategies have been adopted to minimize the damage caused by the drying process, including the adjustment of drying parameters, application of protective agents and induction of cellular adaptation. Gram-positive bacteria, such as LAB, have higher turgor pressure then Gram-negative bacteria due its higher intracellular concentration of potassium pool of amino acids, mainly glutamate [151]. In a hypertonic environment, defence against water intake in osmotic stress occurs mainly by the accumulation of osmoprotectants such as glycine, betaine, carnitine and proline [152,153]. These molecules can accumulate in high concentrations into the cell without affecting the biological activities because they are not ionized under cellular condition [153]. In view of this, the use of osmoprotectants during the lyophilisation process is highly recommended.

Dried probiotic cells can also be produced by applying the high temperature method of atomisation using spray dryer device. This method is more advantageous over lyophilization because it is faster and straight forward. It also consumes less energy and can produce ultrafine dry powder (between 10–150 μm) that prolongs the shelf life of the products [140,154,155,156]. However, the drying temperatures, between 150 °C and 250 °C, can have detrimental effects on the cell components and compromising the product quality if not finely controlled and optimized. The drying process must follow the steps that consist of sample atomization and exposition to the drying gas, evaporation and harvesting of the dry powder [156]. Thus, the characteristics of the final product depend on the equipment design and the parameters established during drying and must be optimised for each probiotic strain [155]. Atomized probiotic bacteria have low water activity, greater stability under luminosity, oxidation and high temperature during storage conditions, while it is easy to handle, to store and to disperse in aqueous solutions [157,158]. However, the exposition of the probiotic to high temperatures can cause thermal, oxidative and osmotic stress to the cells, leading to ribosome impairment, damaging the membrane proteins and destabilization of the membrane structures [159,160].

The effectiveness of any drying processes depends on the retainment of some water in cell cytosol. The excessive loss of water produces intracellular desiccation that induces a cellular state called anhydrobiosis, in which vital functions are paralyzed. So, the challenge is to paralyse but not inactivate the cell metabolism. Alpert [161] defined the state of total desiccation when a cell has less than 10% of its mass composed of free water. The maintenance of cell viability in the anhydrobiosis state depends on the activation of the desiccation tolerance mechanism. There is a difference between the concepts of dryness/dehydration and desiccation [161]. In dryness or dehydration state, the cells have low water activity inside enough to maintain the molecules in their solvation state. On the other hand, the desiccation state is characterized by the low water activity and the lack of the hydration shield of the molecules. This water shield loss mainly affects proteins which play roles vitals to the cells, such as enzymes, regulators and membrane components. In this context, *S. cerevisiae* cells need to produce molecules that will protect the integrity of the proteins, such as trehalose and heat shock chaperones (HSP) [162]. Trehalose was pointed to have the most relevant role in protecting against desiccation by stabilizing macromolecules with the formation of hydrogen bonds and also by vitrifying the cytoplasm to protect cellular components [163]. The process of desiccation can also generate ROS by the Maillard reactions in the cytosol between amino acids and reducing sugars and by the Haber-Weiss/Fenton reactions between ferrous ion and molecular oxygen [164]. These ROS can promote lipid peroxidation and damages to proteins and DNA. Hence, protection against these damages plays a fundamental role for the maintenance of cell viability during and after drying during industrial processing.

Although cellular metabolism is paralysed in the anhydrobiosis state, there is a lot of evidence that some biological functions persist, such as the activities of some enzymes, the expression of some genes and even the generation and use of energy. This maintenance of minimal metabolism allows the cells to restore their biological activities during rehydration. The cellular condition before the process of desiccation is an important factor that helps their recovery after drying. Bacterial cells in stationary phase are more tolerant to desiccation [165]. The stationary phase is known to be marked by a metabolic shift that prepare the cells to face nutrient starvation, activating the General Stress Response (GSR) and the mechanisms that extend cell longevity. Part of GSR involves the diversion of resources primary destined for growth to a set of mechanisms that ensure cell survival and prevent or repair damages to molecules and components [166,167]. This GSR partially overlaps the so-called stringent response, a mechanism that signals the cell entry into the stationary phase due to the scarcity of amino acids in the medium [168]. This nutritional privation induces the production of enzymes involved in amino acid biosynthesis. It is important to note that some amino acids also act on acid stress tolerance [126]. Therefore, the addition of protective molecules in the cultivation substrate in the stage of biomass production and the induction of molecular mechanisms of tolerance should contribute to the successful maintenance of the population of probiotic cells both during drying and during rehydration. The protection of cells in matrices in the form of microcapsules seems to be an important technological alternative to maintain adequate cell viability.

Microencapsulation is a process currently used to maintain cell viability and vitality during drying processes [169]. In addition, this process allows better protection of probiotic bacteria during processing, storage and passage through the GIT to their site of adsorption [139,170]. This process can be performed by physical–chemical or mechanical methods, trapping the cells in structures of up to 1 mm of diameter circumscribed by a permeable, semi-permeable or non-permeable membrane composed by the protective agent used. This is performed during the drying of the cells by lyophilization or atomization [139,171,172]. Besides the protective action, the materials used to encapsulate probiotics have to be safe for consumption, and present emulsifying potential, low viscosity and biodegradable features [173]. Materials used as protective agents include pectin [174], maltodextrin [175], carboxymethylcellulose, sodium alginate [176,177], starch, collagen [178], whey proteins, Arabic gum [179], among others. Therefore, the type of material used as encapsulant, the initial cells density and metabolic state and the size of the particles can interfere with the final viability of probiotic bacteria [180]. Many studies on the increasing probiotic viability after the microencapsulation process can be found in the literature, some of which are resumed below:(a)Salar-Behzadi et al. [181] microencapsulated *B. bifidum* BB-12 using Arabic gum, gelatine and pectin as protective agents, and observed partial protection of the microorganism from membrane damage caused by the drying process. In addition, the product maintained greater viability after one month’s storage.(b)Bora et al. [182] performed microencapsulation by lyophilization of *L. acidophilus* using whey and fructooligosaccharides (FOS) as protectors to enrich lyophilized banana powder. Besides the better protection of the cells, it did not interfere with the sensory characteristics of the banana powder.(c)Nunes et al. [183] used the spray-dryer in association with protective agents such as inulin, hi-maize^®^ and trehalose to microencapsulate *L. acidophilus* LA-5. Trehalose protected the cells from GIT simulated conditions and extended the product stability beyond 120 days of storage. In addition, hi-maize^®^ turned the cells more resistant to heat treatment.(d)Rosolen et al. [184] used spray-dryer, whey and inulin to microencapsulate *L. lactis* subsp. *lactis* R7 and observed a high level of cell protection against simulated GIT conditions and heat treatments.(e)Leylak et al. [185] carried out spray-dryer microencapsulation of *Lactobacillus acidophilus* LA-5 using powdered whey and Arabic gum as protective agents. The results obtained after exposure to simulated GIT conditions showed better survival capacity of the encapsulated microorganisms.(f)Minami et al. [186] used a method based on interfacial tension to produce a three-layered capsule composed by gelatin and pectin in the outmost layer and vegetable fats and oil in the middle layer to encapsulate *B. brevis* and *B. longum* cells. The results showed that *B. brevis* cells not only survived as they also grew when in the GIT of children.(g)Costa et al. [187] showed that encapsulated Bifidobacterium longum 5^1A^ for bacterial protection during atomisation. In addition, the cells were protected from the acidity of acerola pulp. The study concluded that acerola pulp is an interesting matrix for the production of probiotic fruit juice.

These reports show that there are already very promising technologies for the large-scale production of dry and effective probiotics to meet the growing world demand for these products, whether for direct human and animal consumption, or to add to food matrices to produce solid functional foods, such as cheeses, and beverages.

### 2.4. Probiotics Added to Food Matrices: A Tasty Way to Stay Healthy

Dairy matrices are the most used for the incorporation of probiotics since the consumers are more familiar with the microorganisms incorporated into these foods [188]. Furthermore, the therapeutic functions of foods carrying probiotic microorganisms have been well reported in the literature. In Brazil, for example, the probiotic “Minas Frescal” cheese containing *L. acidophilus* provided an improvement in the immune response, preventing infections and in the attenuation of stress in adult Wistar rats submitted to intense physical exercise [189]. This same cheese containing *L. lactis* was able to prevent colitis in mice caused by sodium dextran sulphate, an agent that provokes intestinal inflammation in animal models [190]. The results of that work indicated a reestablishment of the intestinal barrier because of increased gene expression of tight junction proteins, and the release of the anti-inflammatory and immunosuppressive cytokine interleukin 10 (IL-10) in the spleen and lymph nodes [190]. In the context of liquid matrices, milk fermented by *B. bifidum* MF 20/5 was able to inhibit angiotensin-converting enzyme (ACE) via proteolysis, which could cause antihypertensive effects in the host [191]. On the other hand, the regular ingestion of 200 mL of the shake containing the probiotics *L. acidophilus*, *B. bifidum* and the prebiotic FOS for 30 days was able to positively influence glycaemic and lipid levels (total cholesterol and triglycerides) in 20 volunteers [192].

Furthermore, the incorporation of probiotics in foods is technologically interesting because the addition of some organoleptic characteristics such as aroma, flavour and texture (production of exopolysaccharides). However, probiotic viability in food matrices can be reduced by the manufacturing and storage process, food fat content, pH variation, oxygen levels, water activity, presence of inhibitory substances, salt, sugar, flavourings and dyes [135,193,194,195]. The shelf life of probiotic can be increased up to 12 months when the water activity of the product is less than 0.25 [196].

The storage of probiotics incorporated in some foods is performed at low temperatures. However, this process contributes to lessening the membrane fluidity and to interfering with basal biological processes, such as gene expression, protein synthesis and DNA replication, with negative effects on cell viability [197]. The packaging used for food commercialization can also interfere with the viability of probiotics, as they are usually made of plastic material, with the possibility of oxygen permeability [198]. This can cause oxidative stress which is harmful to bacteria such as *Bifidobacterium*. Thus, glass containers are preferable because of their low oxygen permeability. However, this option is more expensive at the same time as it promotes a greater risk of accidents during handling [198].

Therefore, strategies capable of increasing the stability of probiotics in food matrices during processing and storage are necessary. These strategies include the induction of the microorganism’s tolerance to different types of stress during the production stage and microencapsulation with protective agents, inducing stability and providing controlled release into the food matrix [199]. Another promising alternative is the production of probiotics in dry form for later addition to food matrices. This would increase cell viability and vitality while increasing its shelf life.

## 3. The Probiotic’s Passage through GIT: To the Mouth and Beyond

Once produced, processed and stored properly, probiotic cells are ready to be consumed and to perform their beneficial function of supporting the health of humans and other animals. In this discussion, the paraprobiotics will not be included, only the live cells of the probiotics themselves. However, the reader should remain alert to the fact that dead cells also have their functionality, mainly due to the induction of the host’s innate immune response promoted by the cell surface components. In this context, the Surface Layer Protein from *L. casei* fb05 was able to antagonize the effects of *E. coli* and *Salmonella* both by inhibiting the adhesion of pathogens to the epithelial cells and by inhibiting pathogen-induced apoptosis [200]. Living cells, that survived to the processing, transport, and storage, will face the challenge of arriving alive at the final destination in the intestinal mucosa, where they will act directly or through their postbiotics in promoting the beneficial effects. For that, they will have to face a series of physical, chemical and biochemical challenges. The following explanations will be based on the human GIT as a model, but it can also be applied to other animals.

### 3.1. Overviewing the Metabolic Challenges of Probiotic Cells in the GIT

The survivability of any microorganism in nature relay on its capability to acquire the nutrients from the medium to sustain its metabolic processes [166,201]. Unlike the external environments such as soils, lakes and seas, GIT is far richer in nutrients while represents a very challenging roadway for the bacterial cells until its final fixation in the intestinal surface [202,203,204,205]. The microbiome competes for the exploitation of those nutritional resources and this competition shapes the microbial population diversity. On the other hand, the host also compete to harness the compound required for its own nutrition, especially regarding the almost universal assimilable compounds such as glucose or other simple carbohydrate [205]. Probiotic strains can be introduced in the GIT microbial population by direct ingestion of cell suspensions, pills, through alimentation with fermented foods or composed food matrices, as explained above. Once getting there alive, they compose the host microbial population of the intestines. Due to the possibility of being slowly replaced by the host native species by simple competition, continuous intake of these cells is strongly recommended. While remaining in the GIT, the probiotic species can produce metabolites, such as lactic acid and acetic acid, to the host cell from several carbohydrate that also can inhibit pathogens. A diet enrichment with prebiotics can improve the survivability and persistence of the probiotic in the GIT and their interaction can bring many benefits for the treatment of metabolic and infectious diseases [206]. Usually, prebiotics are complex carbohydrates that require a specialized enzymatic arsenal for their hydrolysis to mono and/or oligosaccharide that can entry the CCP of the probiotic metabolism (Figure 2). Once fixed in the CCP, the corresponding monosaccharides suffer biotransformation to end up as biomass, fermentation products, CO_2_ and water through the different types of metabolism (Figure 3). The fact that LAB are unable to respire, and that oxygen can be a growth inhibitor, the oxygen-limited condition of GIT turns as advantage for the probiotic growth in that environment. In that case, fermentation will take place and the cells could produce a series of metabolites that can act as postbiotics by aiding the host health.

Probiotics in GIT need to extract energy in the form of ATP for any accessible molecule such as the carbohydrate and the easiest way to do that is through the CCP. This biochemical energy is further used for anabolic reactions to produce biomass in the GIT and feed different cellular processes such as motion and nutrient transport, as well as to protect the cells from different forms of cellular stressors, as it will be discussed below. Glycogen production and the PPP are essential to ensure the bacterial survival and competitiveness. In *L. acidophilus*, the production of glycogen was described as providing competitive advantage for cell in this environment [118]. LAB mutants impaired in glycogen accumulation are more sensitive to bile stress, one of the main stressors during GIT colonization [117]. In addition to glucose, other monosaccharides such as mannose and galactose can be metabolised by the CCP to produce energy and biomass. These are the hydrolysis products of prebiotic like GOS and mannooligosaccharides (MOS) from vegetables, fruits and the dietary plant fibre hemicellulose [207,208]. The assimilation of GOS and MOS in the diet is reported to improve the survival and competitiveness of probiotic species in the GIT [93,207] or to increase the health effect of probiotics by the production of postbiotic molecules such as short chain fatty acid (SCFA). FOS coming from inulin are also acting as prebiotics to produce fructose, with beneficial activity for the probiotic cells and, consequently, to the host [209,210]. Therefore, a combination of probiotic with plant-rich diet is always advisable.

### 3.2. Oral Mucosa: The First Challenge at the Front Door

The mouth is the first contact of the probiotic with the host. The saliva contains two groups of antibacterial proteins: immunoglobulins that act on the innate immune response and lysozyme, the most important proteins in the prevention of oral infections [211]. Lysozyme is a β-1,4-N-acetylmuramidase that cleaves the glycosidic bond between N-AcGlu and N-AcMu in the peptidoglycan layer of the cell wall, resulting in loss of wall integrity and cell death [212,213] (Figure 4). In addition to its enzymatic activity, a 9-amino acid cationic antimicrobial peptide within the lysozyme structure can induce the formation of pores in the negatively charged bacterial membranes, leading to its disruption [214,215]. Probiotics are mainly composed of Gram-positive bacteria that are naturally more susceptible to lysozyme due to the large amount of peptidoglycan in the cell wall compared to Gram-negative bacteria [216]. However, LAB strains resistant to lysozyme can be selected, such as some strains of probiotic *L. lactis* that show modification of their peptidoglycan by O-acetylation at the C-6 position of N-AcMu, preventing the enzyme from acting (Figure 4) [217,218]. Bacteria that do not have peptidoglycan modifications are lysed. Then, the peptidoglycan fragments are released and detected by host proteins called pattern recognition receptors (PRRs) to trigger the innate immune response in the host [219]. In contrast, peptidoglycan-modified strains are resistant to hydrolysis by lysozyme, and they remain intact to the destination. Therefore, resistance to lysozyme has become a promising criterion for the selection of new probiotic strains [220], although this may to some extent diminish the immunogenic potential of the probiotic. For example, Inayah et al. [221] reported the screening and selection of strains with probiotic potential of *L. acidophilus* resisted 100 mg/L of lysozyme under simulated salivary conditions. In addition to saliva, lysozyme is also found in blood, liver, tears, urine, milk, on mucosal surfaces, and in the gastric juices of mammals [212]. 

This resistance can be a consequence of genetic modifications that include mutations in the genes encoding enzymes of the peptidoglycan construction, but it can also be induced by modulating the expression of these genes by environmental conditions. In a model of study, the treatment of *L. vini* cells with sub-MIC concentrations of organic acids (acetic and lactic) or with HCl promoted a change in the constitution of this component to make it resistant to the action of lysozyme, as observed for cells in the stationary growth phase [123]. The entry into the stationary phase in the case of LAB is characterized by the accumulation of organic acids excreted into the medium [222]. Thus, the cell growth product (lactic and/or acetic acids) itself would be affecting the cell wall and peptidoglycan structure [123]. Increased lysozyme resistance was also seen in *L. casei* cells after exposure to osmotic stress [223]. Therefore, treating bacterial cells with the correct stimulus at the end of the industrialization phase can help to produce a probiotic that is more stable in its peptidoglycan structure and resistant to the action of lysozymes in the oral mucosa.

### 3.3. Stomach: The Acidic Battle Field

After passing through the mouth and oesophagus, the probiotic bacteria reach the stomach and face the low pH of the gastric fluid, which can range from 3.5 (0.3 mM H^+^) to 1.5 (30 mM H^+^) [224]. Gastric juice is a combination of hydrochloric acid (HCl), lipase and pepsin for the initial steps of food digestion and for inactivation of invading microorganisms [225,226]. At low pH, pepsin is activated and can act on peptide portion of the cell wall between the meso-diaminopimelic acid (m-DAP) and L- Alanine [227]. Tokatli et al. [228] reported that *L. brevis* MF343 was sensitive to pepsin, which jeopardises its effectiveness as probiotic. Lipases, on the other hand, can destroy the microorganism’s cell membrane, which is composed mainly by phospholipids and glycolipids [229].

Acid tolerance of LAB has been the subject of several studies [230,231,232] (Figure 5). Acidity is a potent antimicrobial, as it causes protein denaturation, DNA damage, and cytoplasm acidification that interrupts enzymatic reactions and affects membrane potential [233]. Lactobacilli species are considered intrinsically resistant to acids, but gastric juice tolerance profile is species (and strain) specific [234]. After entry into the cells, organic acids dissociate into the H^+^ proton and the corresponding ion (acetate or lactate), which leads to increased intracellular acidity that induce metabolic disorders [235,236]. During acid stress, H^+^ accumulate in the cytoplasm and must be extruded to the exterior. This movement can disrupt the proton motive force (PMF) and produce perturbations to the cell metabolism. PMF is a measure of the energetic state of the cell membrane generated by a separation of charge between the cytoplasm and the external environment and created by the membrane potential and pH gradient across the membrane [237]. In lactobacilli, the PMF-dependent proton efflux pump is one of the most important acid tolerance mechanisms, acting to maintain pH homeostasis [238]. The F_O_F_1_-ATPase proton efflux pump is formed by the F_O_ complex bound to the plasma membrane and the F_1_ complex bound to the cytoplasm [239]. During acid stress, the pump has the function of exporting protons from the cell interior with ATP expenditure [240]. Consequently, increased F_O_F_1_-ATPase activity requires an accumulation of energy to increase the cells’ ability to regulate pH homeostasis. Hence, cells subjected to acid stress expend a lot of energy in the form of ATP to maintain viability. The cell can be metabolically inactivated when the energy charge drops to minimum levels (EC < 0.7), and it may cease to exert probiotic action. However, the dead cells could maintain some proprieties as paraprobiotics.

There are some molecules that can aid cells against cytoplasm acidification. Amino acids such as arginine, glutamate, glutamine and histidine are important protectors against acid stress (Figure 5) [123,241]. This protection occurs either through proton neutralization or ATP production, which drives the PMF-dependent proton efflux pump [242]. Arginine is a known ammonium precursor (a neutralizing agent) via the arginine deiminase (ADI) pathway (Figure 5) [243]. ADI comprises three enzymes: arginine deiminase, ornithine carbamoyltransferase and carbamate kinase, encoded by the *arcA*, *arcB* and *20rc* genes, respectively. Each molecule of arginine through ADI provides two moles of ammonium that acts as H^+^-quenchers and one unit of ATP that can be used in the proton efflux pump [125,239]. Glutamate-dependent acid tolerance involves decarboxylation of glutamate to form gamma-aminobutyric acid (GABA) with the consumption of a proton present in the cytoplasm, and removing it from the intracellular medium [244,245]. In LAB, GABA is exported to extracellular environment and can be used in enrichment of fermented foods with benefits to the host [246,247].

Additional mechanisms for the acid stress response act on protein repair, through the action of chaperones and chaperonins such as GroES [126] and modification of the cell membrane proton permeability [248]. The regulation and control of these mechanisms are exerted by the two-component systems (TCSs) and by alternative sigma factors that respond to acid stress by modulating gene expression [249] (Figure 5). The alternative sigma factor RpoS (σ38), for example, is activated in response to stresses such as UV radiation, acid, temperature, osmotic shock, oxidative stress nutrient deprivation and regulates a general stress response [250]. In *L. vini*, the genes *sigV*, *rpoE* and *rpoN* that encode the alternative transcription factors σ5, σ24 and σ54, respectively, had their expression increased by more than four times by exposure to HCl (0.3 mM H^+^), together with the *uspI* to *uspV* genes that encode the so-called “Universal Stress Proteins” [126]. This pattern of gene induction by acid stress was like that observed after oxidative stress, indicating that these two types of stress might produce the same or similar cellular damage [126]. Acid stress also harms the cell wall structure, inducing its remodelling, and anticipate the entry to the stationary growth phase [123]. Some reports suggest that conditions that demand ATP for cell tolerance to acid or bile stresses triggers the reabsorption of cell wall components for the ultimate production of Fru-6P that goes to the glycolytic pathway for energy production [123,251]. Additionally, the cell wall component reabsorption encompasses the sialic acid pathway that is required for efficient gut colonization of neonate’s animal model due to the presence of oligosaccharides in milk [252].

### 3.4. Intestine at Last: The Destination, but Still Full of Pitfalls

The passage through the stomach takes five to 120 min, and then the probiotic bacteria reach the intestine [108,174]. In the small intestine, pancreatic juice is released into the duodenum. This is a solution with pH between 8.3 and 8.6 that contains pancreatin, a set of enzymes composed of trypsin, amylases, lipases and nucleases. Derrien and Vlieg [253] suggested that pancreatin may have a negative effect on probiotic cells, but so far, its mechanism of action is still unclear. At the same time, the duodenum receives the bile juice, or just bile, which is produced by the liver and stored in the gallbladder. This alkaline liquid is composed of bile acids (or steroid acids), bile salts, bilirubin, cholesterol and phospholipids that play an important role in the emulsification and solubilization of lipids in food [254]. In addition, bile has also antimicrobial properties at values greater than 40 mM, which limits the microbiota present in the intestine [255]. At this point, LAB cells need to be resistant to several enzymes and to bile acids. This last parameter is so important that constitutes one of the key steps in the set of in vitro assays to select probiotic strains.

The buffering of the gastric juice (acid) by the pancreatic and bile juice (alkaline) causes the pH in the intestine to oscillates between 7.5 and 8.5. In general, LAB cells hardly grow at this pH range, and may even lose viability, as in the case of a probiotic *L. rhamnosus* strain [256]. In this environment, this bacterium expresses genes that encode the so-called alkaline shock proteins (Asp) that allow its survival in the abrupt change from acidic to alkaline pH [256] (Figure 6). Other mechanisms are activated by LAB to maintain viability at alkaline pH: (1) active extrusion of potassium and the proton-potassium antiport system, (2) activation of the sodium-proton antiport system, and (3) formation of transmembrane proton gradients (ΔpH) in a reverse direction [257]. The alkalization-dependent K^+^/H^+^ antiport system acts by expelling K^+^ from the inside and importing H^+^ from the extracellular environment (Figure 6). Protons are used for the protonation of unprotonated amines that are taken up by cells at alkaline pH [258]. A similar mechanism occurs in the Na^+^/H^+^ antiport system, in which the intracellular sodium is expelled from the cytoplasm with the entry of H^+^ [259]. Sawatari and Yokota [260] showed that *L. acidophilus* JCM 1132 acidifies the medium when it is adjusted with NaOH, but not when KOH is used, showing the functioning of the Na^+^/H^+^ antiporter.

The main target of bile is the microbial membrane. Exposure to bile modifies lipids in the cell membrane, with changes in cell permeability and in the interactions between the membrane and the external environment. Others bile-induced damages to bacterial cells are reported, such as the induction of oxidative stress and DNA lesions, alterations in sugar metabolism and induction of protein misfolding [261]. Furthermore, the dissociation of bile acids in the bacterial cytoplasm releases protons, causing intracellular acidification [261]. However, some Lactobacilli can use bile acids as environmental signals and, in certain cases, metabolize them from bile hydrolysis to use them as nutrients and electron acceptors [262]. The stress caused by bile is multifactorial and therefore implies a variety of processes aimed at detoxifying bile and neutralizing its deleterious effects on bacterial structures [263]. The most important mechanisms for eliminating bile from the bacterial cell include bile efflux pump, bile acid hydrolases, oxidative stress response, regulation of the glycolytic pathway, general stress response, and chaperone proteins [254] (Figure 6). The ATP-dependent bile efflux pump is present in *L. acidophilus* and *B. breve* and its function is to expel bile to the external environment against the concentration gradient with ATP expenditure. Hydrolases catalyse the hydrolysis reactions that release glycine and taurine from bile acids allowing intestinal bacteria to metabolize the resulting anionic carbon chains [264] (Figure 6). These bile acids can also trigger oxidative stress in bacteria, and it results in damage to nucleic acids, free amino acids or amino acids incorporated into proteins and protein cofactors (Figure 6). ROS-dependent oxidations of DNA nucleobases (e.g., 8-oxo-G) or DNA structure lesions (single-strand breaks) are recognized, and trigger gene expression responses mediated by transcription factors induced by oxidative stress, e.g., OxyR, PerR and SoxR [265]. To protect the cell against these damages, LAB have enzymes such as superoxide dismutase (SOD) to convert superoxide into O_2_ and H_2_O_2_ and peroxidases to remove the produced H_2_O_2_, but many of them are devoid of catalase [266]. Most of these activities require energy in the form of ATP, which increases the metabolic flow via the glycolytic pathway [251,267].

GSR is another important mechanism of bile salt tolerance, which is composed of signal transduction elements, central regulatory elements and a set of genes that encode repairers of biological damage caused by different forms of stress [222]. GSR acts to neutralize the damage caused by bile in cell wall disorganization, oxidative stress, DNA damage, protein denaturation and intracellular acidification [268]. Furthermore, chaperone-like proteins produce proper folding of nascent proteins during stress exposure, while proteases promote the removal of damaged proteins [269]. Bile tolerance is one of the most important properties for probiotic bacteria, as it determines their ability to survive in the small intestine [263].

Upon reaching the intestine viable, probiotic bacteria can temporarily colonize or establish themselves in the microbiota. Intestinal mucosal cells are lined by a glycocalyx composed primarily of glycosylated proteins (mucins), glycolipids, immunoglobulins, and electrolytes, and this physical barrier may protect against microbial colonization [270]. For the establishment of the probiotic in the intestinal microbiota, two mechanisms are fundamental: adhesion to the intestinal epithelium and the quorum sensing system. The adhesion of *Lactobacillus* and *Bifidobacterium* cells to the mucosa depends initially on non-specific physical binding by hydrophobic interactions and later on a second stage of adhesion by specific cell wall components called mucin-binding proteins [271]. Mucin-binding proteins are adhesin proteins with Mub and/or MucBP (MUCin-Binding Protein) domains that are bound to bacterial peptidoglycan by a C-terminal Leu-Pro-any-Thr-Gly(LPxTG) motif, and they are capable of binding to mucins in the intestinal epithelium [272]. Then, a quorum sensing (QS) mechanism acts by communicating between bacteria in order to regulate the gene expression of cells in a population. For the QS mechanism, bacteria produce extracellular signalling molecules known as autoinducers. Gram positive bacteria, like most probiotic strains, produce peptides as signalling molecules detected by the histidine kinase receptor of the two-component system, which auto-phosphorylates and activates the transcription of QS genes for biofilm formation. This biofilm is formed by a high bacterial cell density surrounded by a matrix composed of extracellular polymeric substances that make up exopolysaccharides (EPS), forming a type of hydrogel that surrounds and provides protection to cells [273]. The formation of probiotic biofilms in the healthy intestine prolongs bacterial residence time and therefore promotes the exchange of nutrients between the host and the microbiota [274]. The formation of probiotic biofilms is also important because it prevents the colonization of pathogens [275].

In an overall analysis, during their journey through the GIT the probiotic cells are exposed to the acids present in the stomach environment (pH 2–4), bile and gastric enzymes before they reach the large intestine to colonize and proliferate [222,276,277,278]. The acid stress caused by the passage through the GIT can damage the cell wall and membrane of the probiotic [267]. Furthermore, cytosol acidification damages DNA and proteins, leading to cell death [222,279]. In the production process, biotransformations in fermented foods caused by lactic acid, the primary metabolite generated during the fermentation process of lactic acid bacteria, can negatively influence the physiology of probiotic cells [267,279,280]. The cells must, therefore, dodge all these setbacks, arrive alive to colonize the intestine and finally exercise their probiotic functions.

## 4. The End of the Journey and the Beginning of the Job

After fighting all these constraints, the cells finish their journey by reaching the place where they can promote the expected benefits to the host. As a line of reasoning for this review, we will divide these benefits into two types and analyse some case studies as models that should instigate research on other potentialities: antagonism to pathogens and amelioration of metabolic illness.

It is possible to find a large number of references on the positive effects of probiotic bacteria in the prevention and treatment of GIT infections, especially for livestock animals, and listing these studies would make the text much longer. However, it is worth mentioning some examples. *Lactobacillus reuteri* S5 and *L. rhamnosus* SQ511 were capable to combat *Salmonella enteritidis* by inhibiting some of its essential pathogenic features of mobility and biofilm formation [281,282]. The probiotic strain *Enterococcus faecium* EF137V isolated from artisanal “Coalho” cheese, a very traditional cheese in the semi-arid region in Brazil, showed antimicrobial activity against *Campylobacter jejuni* and *C. coli* and a functional food containing this bacterium could help in the treatment of campylobacteriosis [283]. Besides, many species of *Lactobacillus*, *Bacillus*, *Escherichia coli* Nissle, and *Bifidobacterium* have been reported by their antipathogenic actions against *Campylobacter* in humans and animal livestock, making the probiotic-therapy a natural and efficient way to treat gastroenteritis [284]. Besides the protective actions against bacteria, *B. longum* 5^1A^ and *Weissella paramesenteroides* WpK4 decrease the load of *Giardia lamblia* in experimental model and were proposed for the treatment of giardiasis in humans and animals [285]. There are also some reports on the action of probiotics in relieving physiological problems caused by the action pathogenic agents. For example, Nistor-Cseppento et al. [286] used a randomized controlled trail analysis to conclude that the combination of adequate diet and probiotic use contributed to the treatment of sarcopenia in patients with a history of SARS-CoV-2 infection. In addition, the recent review published by Ren et al. [287] makes a very interesting analysis on how the use of probiotics can alleviate and even counteract the various adverse effects that can emerge from the continued use of medications, including chemotherapeutics, some of which have known toxic effects to organs such as the liver and kidney.

Regarding the metabolic illness, we will present two examples among a plenty of reports in the literature to date. First, the direct action of probiotic cells in the control of cholesterol and second the effect of their produced metabolites (postbiotics) in the control of glycemia. The hypocholesterolemic effect of probiotics has been reported mainly though the following mechanisms: bile deconjugation by the bacterial bile salt hydrolase enzyme (BSH), cholesterol assimilation during bacterial growth, cholesterol adsorption to bacterial cell surface and cholesterol conversion to coprostanol [288]. For the first mechanism, the bacteria cells have to produce BSH enzyme from the *bsh* gene that can be acquired horizontally between different bacteria [289]. This enzyme catalysis the hydrolysis of bile salts conjugated to produce cholic acid and glycine from glycholic acid and cholic acid and taurine from taurocholic acid [290] (Figure 6). At low pH, glycholic acid is in the protonated form and becomes more toxic to the probiotic cells, making the expression of *bsh* gene the most important bacterial protective mechanism [290,291,292]. Hence, the deconjugated bile acids that are less soluble and excreted in the faeces. As consequences, the serum cholesterol level decreases by the activation of de novo synthesis of bile acids from cholesterol or by reducing the cholesterol solubility and its subsequent absorption by the intestinal lumen [290]. Park et al. [293] reported the hypercholesterolemia in pigs fed by 10 days with doses of 7.3Log CFU/50 kg of *L. acidophilus* 43121 or a mixture of *L. casei* and *B. longum*. Afterwards, Jones et al. [294] reported hypercholesterolemic in adult humans subjected to the consumption of yogurt containing 10.5Log CFU of *L. reuteri* NCIMB 30242 twice a day and for 6 weeks. Soon after, it became the first commercial probiotic strain for cholesterol reduction that is ready for the food, beverage and supplement market in USA [295].

Regarding the hyperglycemia, the search for natural products put prebiotics and probiotics in the agenda of the pharmaceutical and food industry for reduction of blood glucose [296]. One explanation for the role of prebiotics and probiotics in glycemic control is that they modify the gut microbiota. This newly installed biological community would absorb more glucose through the production of insulinotropic polypeptides and glucagon-like peptides (GLP1). However, this fact is not always observed [297]. Some probiotic strains produce short-chain organic acids such as acetate (C2), propionate (C3) and butyrate (C4), leading to the secretion of incretin hormones that regulate glucose metabolism [298]. One of these hormones is GLP1, which increases insulin secretion while suppressing glucagon. Thus, there is a delay in gastric emptying and reduction of appetite [299]. Butyrate and propionate can also reduce gluconeogenesis in the liver by decreasing the expression of gluconeogenic enzymes such as glucose-6-phosphatase and phosphoenolpyruvate carboxykinase through activation of the adenosine monophosphate-activated protein kinase pathway [300]. Furthermore, lactic acid produced by LAB can also be converted to acetate and propionate via methylmalonyl-CoA or acrylyl-CoA and then to butyrate via acetyl-CoA, promoting the same result on the gluconeogenesis [301]. Overall, there is no consensus in the current literature on the role of probiotics alone in the control of glucose homeostasis. However, there are some straight evidence that the synbiotic approaches, which includes the combined effects of prebiotics, probiotics and postbiotics, are much more effective in this task of healthcare endeavour. Therefore, more studies aiming the selection of specific strains, the control of its dosage and the time of administration composes a promising field of studies [302]. Probiotics were also reported to induce hypothalamic insulin and leptin resistance. The composition of the intestinal microbiome seems to modulate inflammatory response and metabolic pathways in both peripheral and central tissues, indicating its role in preventing insulin resistance as well as obesity [303]. There is currently a huge list of articles that report the benefits of probiotics and their derivatives in the treatment of different diseases, some of them described in the Section 1.3 above. Besides the classical effects on the protection against pathogens, the growing on the metabolic illness, the use of probiotics for the treatment of psychiatric, neurological and behavioural problems have drawn much attention of the researchers. Based on what has been explained about the metabolism of different probiotic bacteria and in the understanding the biological intricacies of the aetiology of the diseases, we can design experiments and suitable experimental models to test the effectiveness of probiotic therapy.

## 5. Microbiome and Faecal Transplant: The Future

So far, we have treated probiotics as related to a single strain or at most as a consortium of a few strains. However, one must consider the microbial complexity of the GIT, especially the intestine, which characterizes the so-called microbiome. This microbiome may be characteristic of the population due to dietary and hygiene acts of the host, but it also presents a composition that is almost specific to each individual. Recent studies have shown the importance of the individual’s microbiome in the origin and/or intensity of different metabolic disorders, such as liver cirrhosis and hepatocarcinomas [304], progression of neurodegenerative diseases [304,305], glycolipids metabolism disorders [306], obesity [307], myocardinal fibrosis and some cardiovascular diseases [308], arterial hypertension [307], among others. These are consequences of the so-called gut dysbiosis, or dysbacteriosis, caused by an imbalance in the composition of the microbiome, altering its metabolic activity as a whole [202,309,310]. For most of the health problems reported herein, a continuous consumption of single- or multi-strain probiotic product as liquid suspensions, powder, fermented products or supplemented solid foods can help to maintain the microbiome and to prevent and restore slight dysbiosis. However, the current perception is that shock treatment with the introduction of a healthy microbiome through faecal transplantation is more effective in restoring the balance of the intestinal microbiota and more rapidly combating metabolic disorders inside and outside the GIT whenever the urgency of the therapy is required [311]. A more detailed description of this topic would be outside the scope of this review and interested readers are encouraged to consult the cited references and the current literature. What we did was to present some very specific studies with probiotics, whose biological mechanisms and action, and not just the phenomenon, have been reported. It is up to the interested reader to choose one of these themes that already being studied, or others not yet explored and to pursue a line of investigation. Thus, we can increasingly expand the exploration of these bacteria as natural agents to promote the improvement of people’s health and quality of life. Additionally, then, “live long and prosper”.

## Figures and Tables

**Figure 1 microorganisms-11-00095-f001:**
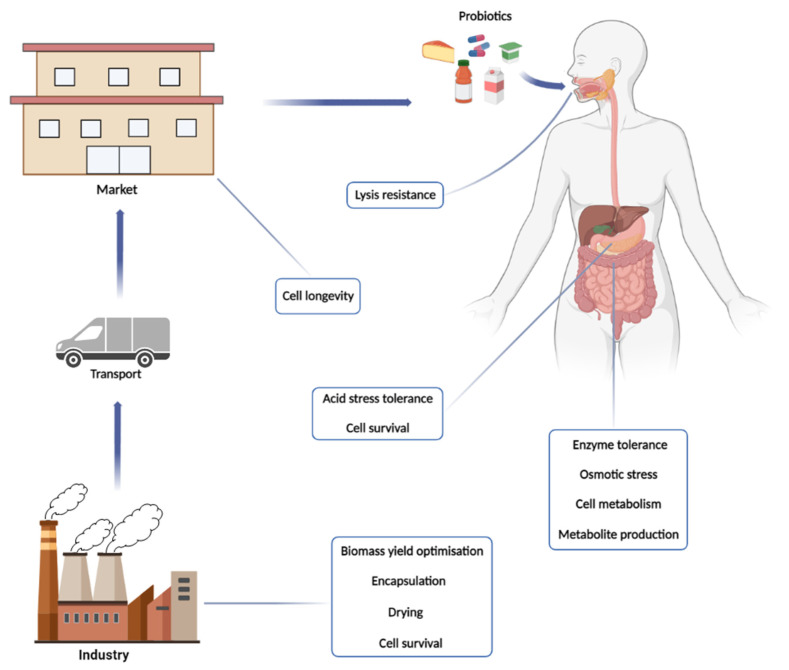
Journey of probiotic products from their production in the industry to their arrival in the intestine, where they will perform their beneficial functions for consumers. In this path, the main challenges and activities of the cells are highlighted.

**Figure 2 microorganisms-11-00095-f002:**
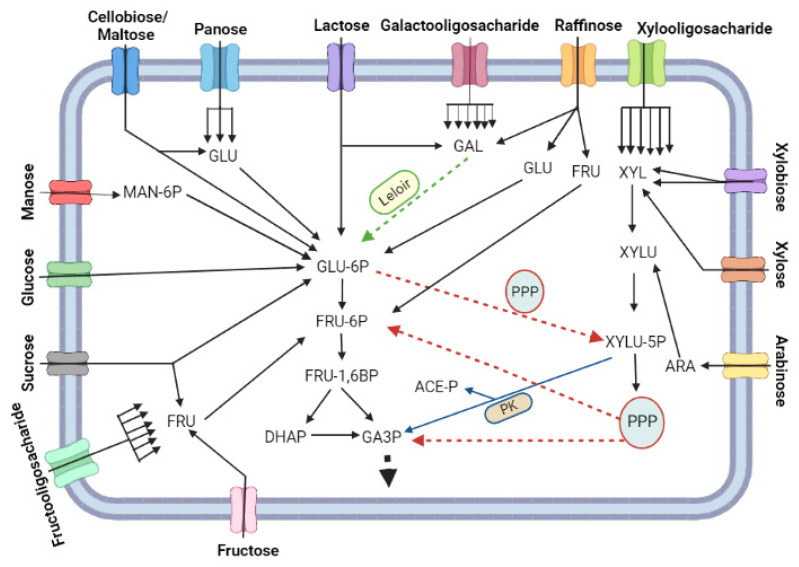
Overview of consumption of different types of monosaccharide sugars (glucose, fructose, galactose, mannose, arabinose and xylose), disaccharides (sucrose, lactose, maltose, cellobiose and xylobiose), trisaccharides (raffinose and panose) and oligosaccharides (fructooligosaccharide, galactooligosaccharide and xylooligosaccharide) and their integration into the central carbon metabolism of probiotic bacteria. Sugars composed of more than one unit must be hydrolysed during passage through transporters or by internal glycohydrolases to release the constituent monosaccharides. Hexoses are converted to glucose 6-phosphate (GLU-6P) or fructose 6-phosphate (FRU-6P). The exception is galactose, which initially must be metabolized by the Leloir pathway. Next, GLU-6P and FRU-6P are metabolized via the glycolytic pathway. GLU-6P can also be metabolized via the pentose phosphate pathway (PPP) to subsequently return to the glycolytic pathway. Pentoses, in turn, are converted directly by PPP. The intermediate xylulose 5-phosphate (XYLU-5P) from GLU-6P or pentoses can be broken down into a molecule of glyceraldehyde 3-phosphate (GA3P) and a molecule of acetyl phosphate (ACE-P) by the enzyme phosphoketolase (PK).

**Figure 3 microorganisms-11-00095-f003:**
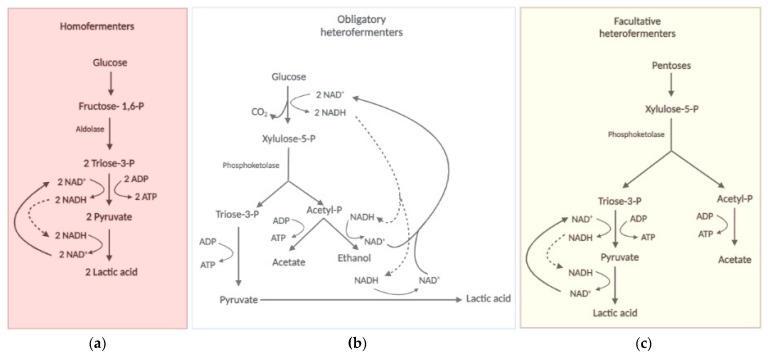
Types of fermentative metabolism displayed by probiotic bacteria. Exclusive homofermenter bacteria utilize glucose or other hexoses exclusively via the glycolytic pathway (**a**). Obligate heterofermenters utilize the PPP pathway and phosphoketolase activity (PPP-PK) to metabolize hexoses and pentoses (**b**). Facultative heterofermenters use the glycolytic pathway to ferment hexoses and the PPP-PK pathway to ferment pentoses (**c**).

**Figure 4 microorganisms-11-00095-f004:**
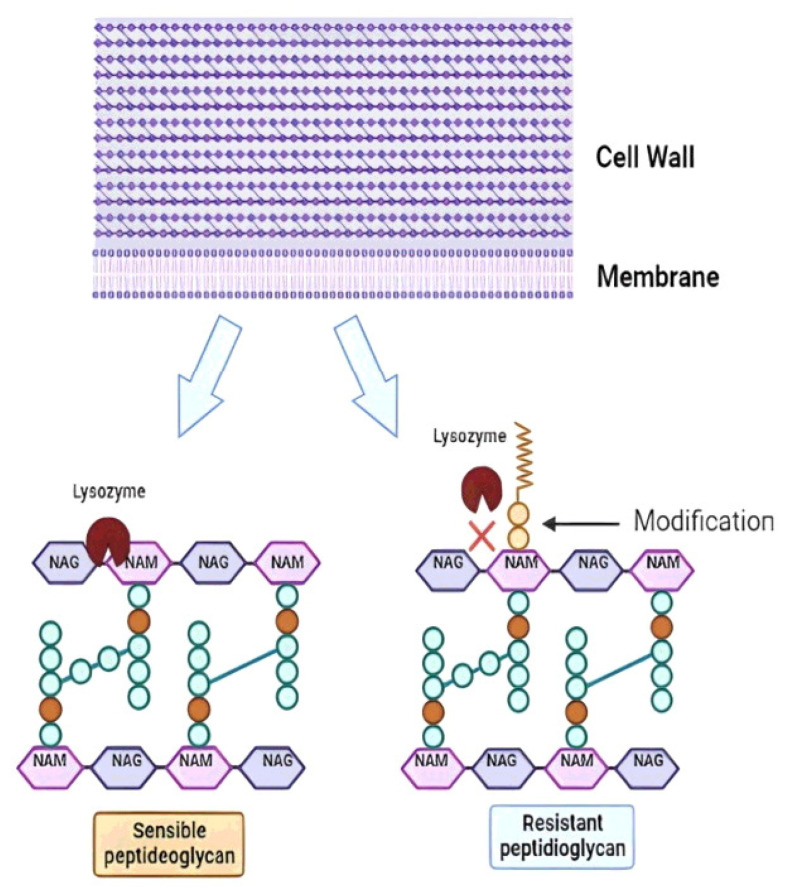
Action of lysozyme on the structure of the peptidoglycan component of the cell wall of probiotic bacteria. In sensitive strains, the enzyme hydrolyses the bond between N-acetylglucosamine (NAG) and N-Acetylmuramic acid (NAM). In resistant strains, NAM residues undergo O-acetylation at the C-6 position. This makes the NAG-NAM bond inaccessible to lysozyme.

**Figure 5 microorganisms-11-00095-f005:**
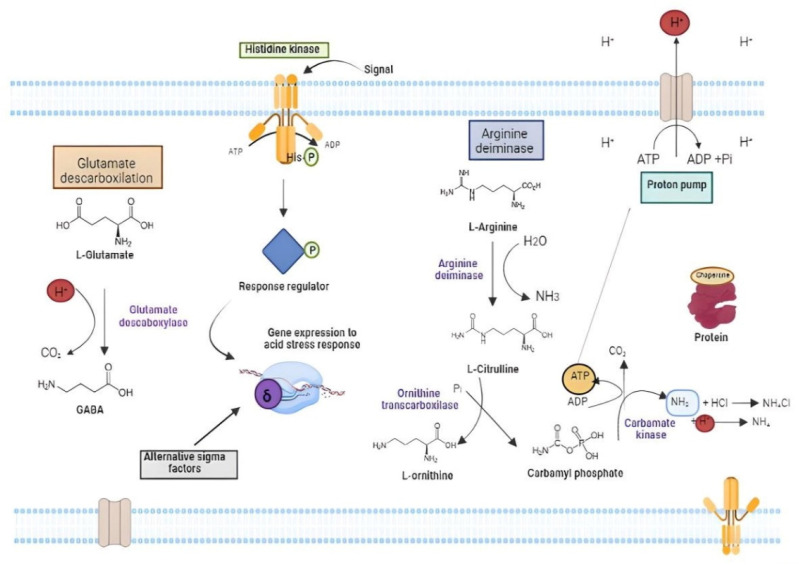
Overview of the main mechanisms of acid stress tolerance in lactic acid bacteria involving amino acid catabolism, expression of stress response metabolism genes by sigma alternative transcription factors activated by histidine kinase and activation of the ATP-dependent proton extrusion pump.

**Figure 6 microorganisms-11-00095-f006:**
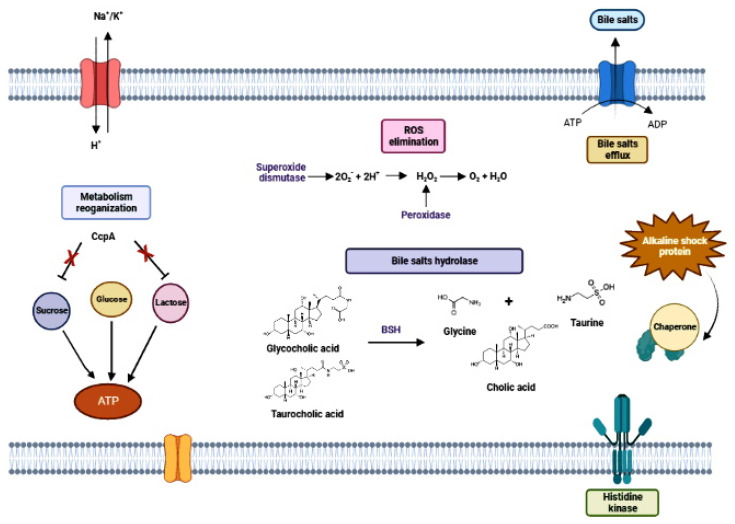
Overview of the main mechanisms of tolerance to intestinal tract stress conditions in lactic acid bacteria from the decomposition of reactive oxygen species (ROS), hydrolysis of bile salts and their excretion by extrusion pumps, antiport ion transport pumps and activation alkaline shock response proteins.

**Table 1 microorganisms-11-00095-t001:** Evolution of the lato sensu meaning of probiotic.

Definition	References
Substances produced by microorganisms that promote the growth of other microorganisms.	[7]
“Organisms and substances which contribute to intestinal microbial balance.”	[8]
“A live microbial feed supplement that beneficially affects the host animal by improving its intestinal microbial balance.”	[9]
“Living micro-organisms administered in a sufficient number to survive in the intestinal ecosystem. They must have a positive effect on the host.”	[10]
“A mono or mixed culture of living microorganisms that benefit humans or animals by improving the properties of the indigenous microflora.”	[11]
“Living micro-organisms, which upon ingestion in certain numbers, exert health benefits beyond inherent basic nutrition.”	[12]
“Live microorganisms which when administered in adequate amounts confer a health benefit on the host.”	[13]
“Live microorganisms that are intended to have health benefits when consumed or applied to the body.”	[14]
“Viable or inviable microbial cell (vegetative or spore; intact or broken) that is potentially healthful to the host.”	[15]

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
