# Peer review of "Journey of the Probiotic Bacteria: Survival of the Fittest"

_microorganisms, 2022, doi:10.3390/microorganisms11010095_

Round 1
Reviewer 1 Report
A review by Allyson Andrade Mendonça et al. made a very good impression on me. It is structured, logically and consistently presented, it covers some modern issues that are not covered in other reviews. That’s why I believe that the reviewed article is of interest to the readers of “Microorganisms”.
Minor:
P. 1, line 41: Ellie Metchnikoff → Elie
Table 3 is not necessary in the present form. Such information can be presented in the text. Bulgaricus, → bulgaricus
Numerous grammar mistakes and misprints. E.g.
line 98: the use those → the use of those
line 104: and promote some → and promotes some
line 105: those strains has been isolated from → have been
line 119: probiotics to vaginal use [33]. → for vaginal use
line 147: proprieties
lines 151, 158: point after “et al”
line 250: can be achieve by
line 255-256: more like → more likely
line 256: 37 space °C.
line 274: to stablish → establish
line 278: may represents
and many others. Editing of English language and style throughout the manuscript are required.
Author Response
Dear reviewer, thank you very much for your feedback on our review. In fact, we tried to cover in a sequence of events all the relevant aspects about the use of probiotic bacteria, merging questions related to the productive processes and their biological effects with questions about the metabolism and physiology of this bacteria relevant to each step.
All suggested text changes were accepted and are in red for better identification. Table 3 was disconsidered as suggested, as indicated by the strikethrough text, and the information transfered to the text. But, it was kept in the text for removal by the technical staff of the journal.
We hope that these changes will be sufficient to warrant publication of the ms.
Reviewer 2 Report
A very important paper summarizing data on probiotic selection, characterization and production. However, the paper is too long and overloaded with information unrelated to the Abstract content. The reviewer suggestion is to strictly limit this ms to chapter 2 and to consider writing a separate paper based on actual chapter 3. The rest can be neglected.
More specific comments:
The authors’ considerations related to probiotic – host interactions are not fully supported by sound scientific basis. Same is true for Chapter 1.3 which contains rather chaotically selected data. Generally, the quality of the paper would be much better higher after cutting many unnecessary or obvious information off. The reviewer recommends limiting the paper content exactly to the frames showed in Abstract. The origin of this ms seems to be an academic thesis with all typical commitments, expectations and limitations.
Author Response
Dear reviewer, thank you very much for your feedback on our review. Below are responses to comments and suggestions made. We hope they are sufficient for the ms to be considered suitable for publication.
1) A very important paper summarizing data on probiotic selection, characterization and production.
RESPONSE: In fact, we tried to cover in a sequence of events all the relevant aspects about the use of probiotic bacteria, merging questions related to the productive processes and their biological effects with questions about the metabolism and physiology of this bacteria relevant to each step.
2) However, the paper is too long and overloaded with information unrelated to the Abstract content.
RESPONSE: below we present a relation between the original text in the abstract (between quotes) and the sections in the manuscript (in italic). A brief introduction on the topic of the microbiome was presented in section 5, more to arouse the reader's curiosity about this very current topic and great relevance on how probiotics already fully adapted to the intestine will be useful for emergency and clinical urgency.
“This review aims to bring a more general view of the technological and biological challenges regarding production and use of probiotic bacteria in promoting human health” (section 1.3 and 1.4).
“After a brief description of the current concepts (sections 1.1 and 1.2)…”
“the challenges for the production at an industrial level are presented from the physiology of the central metabolism to the ability to face the main forms of stress in the industrial process.” (sections 2.1 and 2.2)
“Once produced, these cells are processed to be commercialized in suspension or dried forms or added to food matrices. At this stage, the maintenance of cell viability and vitality is of paramount for the quality of the product. Powder products requires the development of strategies that ensure the integrity of components and cellular functions that allow complete recovery of cells at the time of consumption.” (sections 2.3 and 2.4)
“And finally, once consumed, probiotic cells must face a very powerful set of physicochemical mechanisms within the body, which include enzymes, antibacterial molecules and sudden changes in pH.” (sections 3 and 4)
“Understanding the action of these agents and the induction of cellular tolerance mechanisms are fundamental for the selection of increasingly efficient strains in order to survive from production to colonization of the intestinal tract and to promote the desired health benefits.” (sections 1 to 4)
3) The reviewer suggestion is to strictly limit this ms to chapter 2 and to consider writing a separate paper based on actual chapter 3. The rest can be neglected.
RESPONSE: The text is actually a bit long. However, the removal of any of the sections, especially #3, would break the entire design of the text, prepared so that the reader has a complete view of the subject of probiotics, from its conception to its clinical benefits. To our knowledge, no other review has been written with such an overview. Therefore, even respecting the reviewer's indication, we prefer to keep the structure of the text as it is.
4) More specific comments:
4a) The authors’ considerations related to probiotic – host interactions are not fully supported by sound scientific basis.
RESPONSE: sorry, but we completely disagree with this observation. The article has 311 references, of which 63 are related to the bacteria-host interaction. For example, section 4 refers to the effects of bacteria on the host gut, with 83 lines containing 22 references.
4b) Same is true for Chapter 1.3 which contains rather chaotically selected data.
RESPONSE: sorry, but the term chaotic doesn't seem appropriate to us. This section sets out to present a series of case studies showing the beneficial effects of probiotics. Unlike a metabolic analysis, in which events are interconnected and in a biological order, case studies (as the term implies) do not present cause-and-effect relationships. The only relationship here, and even then not mandatory, would be the reference of cases in chronological order.
4c) Generally, the quality of the paper would be much better higher after cutting many unnecessary or obvious information off.
RESPONSE: In reviews there are no such things as "obvious information". All information is important precisely because a large part of the interested public is non-specialists.
4d) The reviewer recommends limiting the paper content exactly to the frames showed in Abstract.
RESPONSE: For the reasons explained in the responses to comments 1, 3, 4a and 4c above, we request that the reviewer reconsider the request for shortening the text
4e) The origin of this ms seems to be an academic thesis with all typical commitments, expectations and limitations.
RESPONSE: This original work was written by academics with different expertise and is the result of several ongoing projects by our research group: a postdoctoral project on the isolation and physiological characterization of lactobacilli from, a doctoral thesis on the production of probiotics from of milk-alternative matrices, a doctoral thesis on the molecular biology of the stress response in lactobacillus and a master's work on the optimization of the drying processes of probiotic bacteria. We think this validates the academic quality of the text.